# Mineral dissolution and reprecipitation mediated by an amorphous phase

Matthias Konrad-Schmolke [1], Ralf Halama [2], Richard Wirth[3], Aurélien Thomen[4], Nico Klitscher[3], Luiz Morales[5], Anja Schreiber[3] & Franziska D.H. Wilke[3]

Fluid-mediated mineral dissolution and reprecipitation processes are the most common mineral reaction mechanism in the solid Earth and are fundamental for the Earth's internal dynamics. Element exchange during such mineral reactions is commonly thought to occur via aqueous solutions with the mineral solubility in the coexisting fluid being a rate limiting factor. Here we show in high-pressure/low temperature rocks that element transfer during mineral dissolution and reprecipitation can occur in an alkali-Al–Si-rich amorphous material that forms directly by depolymerization of the crystal lattice and is thermodynamically decoupled from aqueous solutions. Depolymerization starts along grain boundaries and crystal lattice defects that serve as element exchange pathways and are sites of porosity formation. The resulting amorphous material occupies large volumes in an interconnected porosity network. Precipitation of product minerals occurs directly by repolymerization of the amorphous material at the product surface. This mechanism allows for significantly higher element transport and mineral reaction rates than aqueous solutions with major implications for the role of mineral reactions in the dynamic Earth.

[1] Earth Science Department, University of Gothenburg, Gothenburg 40530, Sweden. [2] School of Geography, Geology and the Environment, Keele University, Keele ST5 5BG, UK. [3] GeoForschungsZentrum Potsdam (GFZ), Telegrafenberg, Potsdam 14473, Germany. [4] Infrastructure for Chemical Imaging at the Chalmers University of Technology and University of Gothenburg, Gothenburg 412 96, Sweden. [5] Eidgenössische Technische Hochschule (ETH), Zürich 8092, Switzerland. Correspondence and requests for materials should be addressed to M.K-S. (email: mks@gvc.gu.se)

Fluid-mediated mineral dissolution and reprecipitation is a key mechanism for mineral reactions in different fields of Earth and material science[1–4]. Mineral reactions in the solid Earth constrain many geodynamic and geochemical processes, such as density changes in subducted oceanic crust[5,6], carbon dioxide release in subduction zones[7], and the precipitation of ore minerals from percolating fluids[8]. Detailed knowledge about rate-limiting mechanisms during mineral reactions is fundamental for the quantification of rock transformations and the associated element transport.

The classical approach to fluid-mediated mineral dissolution and reprecipitation assumes that minerals in contact with a fluid phase dissolve until the chemical potential of the solid phase equals that of the dissolved material[9]. Dissolution of the solid in the coexisting fluid occurs predominantly by aqueous metal hydrolysis and metal-proton exchange reactions[10]. Hydrolysis affects mostly the outermost charged monolayers of the crystal surface[11,12], but in case of metal-proton exchange can penetrate up to several hundreds of nanometers into the crystal surface[10,13]. Once dissolved elements are transported within the fluid over distances ranging from several nanometers to kilometers[14–16]. In turn, precipitation of new phases occurs as a result of super-saturation of the fluid with respect to the product phases. After nucleation of a product, hydrolyzed atoms or molecules from the solution are incorporated into the crystallizing phase at energetically favorable sites at the outermost surface of the precipitating crystal[17]. The surface kinetics of mineral dissolution and precipitation[3,18], the solubility of the reacting minerals in the co-existing fluid[19,20], and the intercrystalline transport rate of the solutes[21–23] are therefore rate limiting for mineral reactions. Regarding mineral reactions in the solid Earth efforts have been made to quantify these parameters but especially the feedback among them is not yet fully understood[2].

Several experimental and natural examples show, however, that dissolution–reprecipitation processes can involve the formation of a silica-rich amorphous phase. The conditions under which amorphous phases are observed as reaction products range from weathering at ambient temperatures[24–27] to metamorphic conditions[28,29]. The role of these amorphous phases as element transport agents and pre-nucleation products are not fully resolved. Several recent publications have demonstrated that amorphous and nano-crystalline precursor phases play a funda-mental role in natural precipitation mechanism, such as the formation of calcium carbonates[30] and oxides[31] from seawater, but an entire mineral reaction mechanism that involves formation of an amorphous phase, element transport therein, and direct precipitation from the amorphous material has never been demonstrated.

Here, we show in situ that naturally occurring mineral reactions in metamorphic rocks can deviate from the classical dissolution-transport-precipitation path. We use a combination of back-scattered electron (BSE) imaging, high-resolution trans-mission electron microscopy (HR-TEM), focused ion beam (FIB) sectioning, and Nano-secondary ion mass spectrometer (SIMS) mappings to visualize dissolution of primary minerals resulting in the formation of solid amorphous material, element transport via the amorphous state in an interconnected, syn-metamorphic pore space, and precipitation of product phases by repolymerization of the amorphous material. Our observations demonstrate that the kinetics of these reactions is not limited by the solubility of the reactant in the fluid, as an amorphous solid material is formed directly from the depolymerized mineral. Precipitation of the crystalline product by repolymerization of the amorphous material and the existence of an interconnected syn-reactive pore space further enhance mineral reaction rates.

## Results

**Hydration reaction during eclogite–blueschist transformation.** The reaction processes are observed in high-pressure (~1 GPa)/low-temperature (~500 °C) rocks that underwent hydration in an ancient subduction zone that is now exposed in the Franciscan Complex[32–34] along the Pacific coast of California (USA). Samples were taken from an ~1 × 1 × 1 m sized block (Fig. 1a) near Jenner north of the mouth of the Russian River (Supplementary Note 1 and Supplementary Figure 1). The sampled block is a scraped-off part from the subducting plate incorporated into a

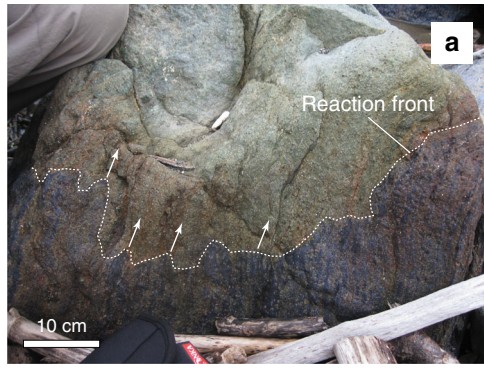

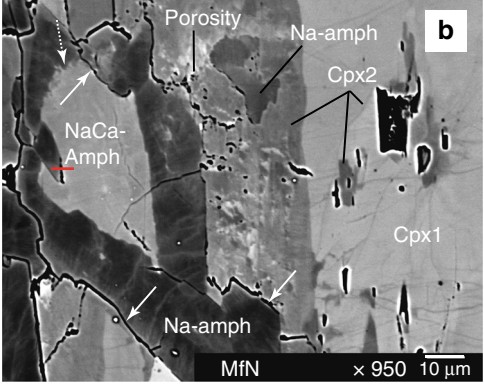

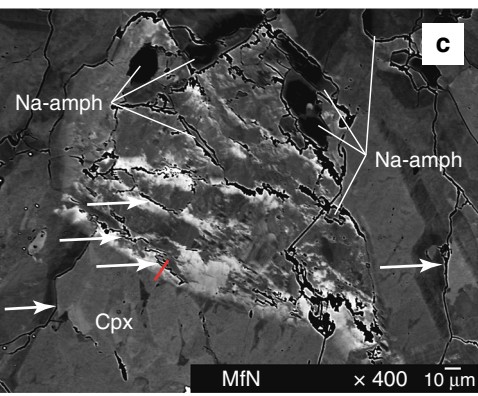

**Fig. 1** Meso- and micro-scale textures resulting from fluid infiltration into the sampled rock. **a** A sharp reaction front in the meter-sized block marks the extent of fluid infiltration (bluish areas) into the dry eclogite (greenish areas). **b** Back scattered electron (BSE) image of a fluid-induced reaction zone. Initial fluid influx in porous areas modifies the clinopyroxene (Cpx) composition (patchy areas). Subsequent formation of sodic amphibole (Na-amph) from sodic-calcic amphibole (NaCa-amph) and cpx is associated with up to 1 μm wide reaction zones (solid arrows). **c** BSE image of the fluid-induced reaction zones in cpx showing the patchy zoning (white cloudy areas) and the associated interconnected pore space together with the formation of sodic amphibole. The red lines mark the positions of the FIB sections from Figs. 2 and 3 taken across the reaction zones

mélange of rocks at the slab-mantle interface along which it was exhumed and partly affected by intense fluid–rock interaction (Supplementary Note 2 and Supplementary Figure 2). The block consists of two parts: a predominantly dry upper part consisting of the high-pressure (eclogitic) mineral assemblage clinopyroxene + garnet + rutile with subordinate sodic-calcic amphibole (NaCa-amphibole) and epidote and a hydrated medium-pressure (blueschist) lower part that contains sodic amphibole (Na-amphibole) + titanite ± lawsonite ± white mica. Textural overprinting relations clearly demonstrate that the eclogitic (greenish) part is affected by a fluid influx causing intense hydration of the primary mineral assemblage. The hydrated (bluish) part is clearly separated by an irregular reaction front (Fig. 1a), which is sharp down to mm-scale (Supplementary Figure 3). Following the pressure–temperature evolution proposed by Krogh et al.[34], this rock transformation is corresponding to decompression and cooling from eclogite to lawsonite-blueschist facies associated with a significant fluid influx. It is notable that the re-hydration of eclogites and the formation of hydrous amphibole from pyroxene is a commonly observed fluid–rock interaction in high-pressure rocks[35]. The reaction is associated with a significant positive volume change and the consumption of fluids percolating in and above subducted plates. Hence, it has crucial implications for the seismicity and melt production in subduction zones[35,36].

BSE images from our samples reveal the complex chemical and mineralogical changes during the hydration reactions. First indications of a fluid influx are given by a network of brittle fractures that affect large, inclusion-rich (rutile and fluid inclusions) primary clinopyroxene grains (Fig. 1b, Supplementary Figure 4). The fractures are filled with secondary clinopyroxene that also occurs as overgrowth on primary clinopyroxene along the grain boundaries (Supplementary Note 3 and Supplementary Figure 5). Secondary clinopyroxene is often associated with pores that sometimes form an interconnected porosity (arrows in Fig. 1c) connecting grain interiors with the grain boundaries (Supplementary Figures 6 and 7). Along this interconnected porosity fluid infiltration induces a chemical modification of the primary clinopyroxene grains, which is characterized by a diffuse, cloudy and patchy appearance of the affected regions in the BSE images (Fig. 1b, c). These cloudy areas emanate from grain boundaries and porous areas and reach well into the interior of the clinopyroxene grains, sometimes with a sharp boundary towards the unaffected areas (Fig. 1c).

Followed by the compositional modification of the clinopyroxene grains is the fluid-induced formation of Na-amphibole. Na-amphibole growth starts in the cloudy and porous areas and consumes all clinopyroxene generations as well as the sodic-calcic amphiboles (Fig. 1b, c, Supplementary Note 4 and Supplementary Figure 6). The reaction front between clinopyroxene and Na-amphibole is marked by porosity that is separating reactants and products (solid arrows in Fig. 1b). In reaction zones between the two amphibole generations the pore space is sometimes missing and the contact between the phases is blurred (stippled arrow in Fig. 1b). The newly formed Na-amphibole exhibits complex internal compositional variations similar to the fluid-affected clinopyroxene grains. In the BSE images, Na-amphibole grains have brighter rims around darker cores the latter of which are truncated by numerous, diffuse and finger-like brighter areas. These brighter fingers connect Na-amphibole interiors, as well as the reaction surface between NaCa-amphibole and Na-amphibole in Fig. 1b, with the grain boundary network.

**Formation of amorphous phases**. We sampled the reaction zones indicated in Fig. 1b, c using a FIB system, to produce electron transparent slices, which were then investigated by HR-

TEM. A bright-field TEM image of the FIB section across the NaCa-amphibole–Na-amphibole reaction zone (Fig. 2a) shows an elongated pore (white) with a maximum width of ~300 nm between the reacting NaCa-amphibole and the crystallizing Na-amphibole. In the lower part the Na-amphibole crystal is in contact with the reactant, which is characteristic for interface-coupled dissolution–reprecipitation reactions[2]. Both crystals have a high dislocation density marked by the dark line contrasts within the grains. The surface of the reacting NaCa-amphibole within the pore is highly irregular, marking its dissolution. The pore space between the mineral grains is occupied by a light gray, cloudy material without any diffraction contrast, thus indicating an amorphous state. This material emerges from the dissolving crystal and reaches the Na-amphibole surface. In contrast to the reacting NaCa-amphibole crystal, the interface of the newly formed Na-amphibole grain is sharp. The HR-TEM lattice fringe image of the Na-amphibole grain boundary (Fig. 2b) demonstrates that the pore filling material is amorphous (lower diffraction pattern) and in contact with the crystalline Na-amphibole. In a transition zone, a few nm above the sodic amphibole surface, the amorphous material starts to form clusters within which the orientation of the Na-amphibole crystal ((010) lattice fringes) can already be seen. The intensity profile along the line across the (010) lattice fringes displayed in the lattice fringe image documents this ordering process in the transition zone (Fig. 2b). The arrangement of and the distances between the lattice fringes (1.7 nm (010)) are irregular on the left side of the profile, they start to arrange in chains and approach a uniform distance between 1.5 and 2.0 nm in the transition zone and are fully crystalline with a lattice fringe spacing of 1.7 nm on the right side of the profile. The latter value is close to the expected (010) lattice fringe spacing of 1.73 nm for the glaucophane (an end-member Na-amphibole) molecule.

A close up high-angle annular dark-field (HAADF) image on the darker defect-rich areas in the NaCa-amphibole crystal shows that the formation of the amorphous material starts along the defect lines and cleavage planes ({110};{010}) within the grain (Fig. 2c). These defect lines are also connected with the formation of nano-scale porosity parallel to the preferred cleavage of the amphibole crystal. It is notable that such defect-rich areas correspond well with areas of cloudy and patchy compositional zoning in both amphiboles and pyroxenes (Supplementary Figure 10), indicating that these linearly arranged dislocations serve with their dislocation cores as element exchange pathways during fluid influx. This interpretation is supported by the observation that the patchy clinopyroxene areas visible in the BSE images are enriched in hydrogen compared to the unaffected parts of the crystals. The Nano-SIMS isotope mappings in Fig. 3 show a clear correlation between the hydrogen content of the nominally anhydrous clinopyroxene (measured as $^1H^{28}Si$ and $^1H^{16}O$ and calculated as $^1H^{28}Si/^{28}Si$ and $^1H^{16}O/^{16}O$) and the bright areas in the BSE image, which in turn correspond with areas of high dislocation density. The correlation between elevated hydrogen concentrations and the high defect densities in the affected clinopyroxene areas support the interpretation that crystal defect cores serve as fluid and element exchange pathways and thus enable in situ transformation of the reacting mineral.

The dissolution of reacting clinopyroxene is likewise associated with the formation of amorphous material although its generation is different from the mechanism in the amphiboles (Supplementary Note 5). Dissolving clinopyroxene in the reaction zones develops needle-like denticulated surfaces resulting from dissolution along preferred crystallographic directions (Fig. 4a, Supplementary Figure 8). The denticles become fringes with diameters of only a few nm and grade into a mesh of amorphous material, which is, as in the case of the amphibole-amphibole reaction, in

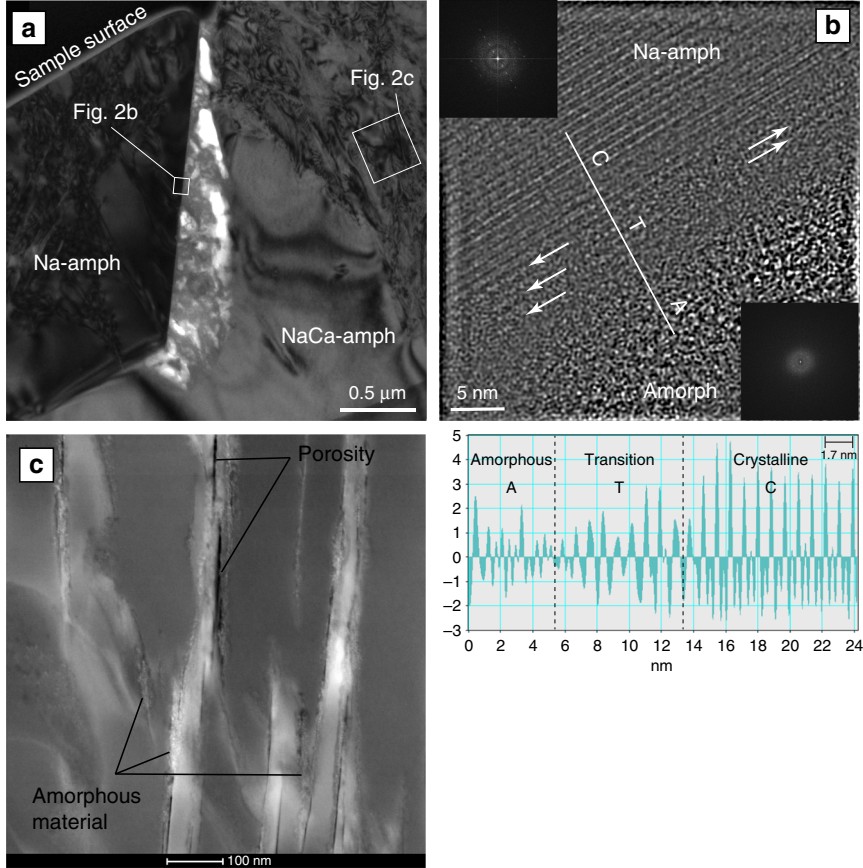

**Fig. 2** Details of the FIB section across the NaCa-amph–Na-amph reaction zone. **a** Bright-field TEM image of the reaction zone in Fig. 1b showing the syn-reactive pore space (white areas) filled with an amorphous material resulting from the dissolution of the sodic-calcic amphibole. **b** Lattice fringe image of the sodic amphibole surface showing the formation of re-polymerized silicate chains from the amorphous phase (Amorph) at the crystal surface. The intensity profile along the white line demonstrates the formation of crystalline sodic amphibole directly from the amorphous material. **c** High-angle annular dark-field (HAADF) TEM image of the dislocation-rich area in **a** showing the initial formation of the amorphous material and porosity along dislocation lines

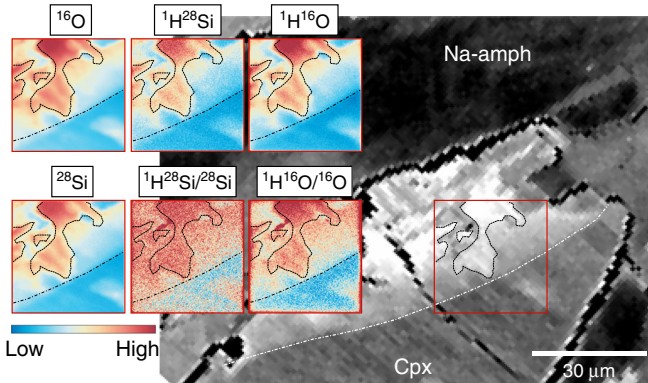

**Fig. 3** BSE image and Nano-SIMS measurements of a clinopyroxene crystal affected by the fluid influx and surrounded by newly formed sodic amphibole. The brighter areas in the BSE image are marked by a sharp reaction front (dash-dotted line) behind which the cpx crystal has a pathy zoning. The reaction front and the patchy areas (stippled outlines) are clearly reflected in the isotope images. The bright areas in the BSE image correspond with elevated hydrogen contents (indicated by the $^1H^{28}Si/^{28}Si$ and $^1H^{16}O/^{16}O$ mappings) in the nominally anhydrous cpx. Clinopyroxene is defined as Cpx and sodic amphibole as Na-amph

contact with the newly formed sodic amphibole. An inverse FFT HR-TEM image (filtering the diffuse scattering intensity from the diffraction pattern) of the contact area between the amorphous material and the denticles in the dissolving clinopyroxene (Fig. 4b) shows a highly lobate and irregular contact between the crystalline denticle, occupying the lower part of the image, and the amorphous material in the upper part. The latter contains nm-sized volume fractions of the dissolved crystal (arrows in Fig. 4b). The calculated diffraction patterns (FFT) on the right side of the image shows the difference between the crystalline clinopyroxene and the amorphous pore filling material. Apart from the highly irregular clinopyroxene surface there are a number of small areas within the crystal where the crystal lattice is modified and amorphous regions emerge within the clinopyroxene (outlined regions in Fig. 4b). These regions can be found up to several tens of nm within the crystal and are between 1 and 20 nm in size. They reflect a continuous transformation of the clinopyroxene crystal into an amorphous state. As in case of the amphibole disintegration the clinopyroxene-sodic amphibole reaction is associated with a syn-metamorphic pore space (white areas in Fig. 4a) within which the amorphous material is in contact with both reactant and product. It is notable that in some pores the amorphous material occurs as µm thick precipitate at

the inner walls of the pore enclosing the denticles of the pyroxenes indicating that the amorphous material can move freely within the pore space (Supplementary Figure 9).

Crucial for the reaction and element transport rate during the observed rock transformation is the composition of the amorphous material. Two energy dispersive X-ray spectra for both types of amorphous material are given in Fig. 5. The composition of the amorphous material from amphibole (a) and clinopyroxene (b) differ in their Al and Ca contents, but reflect the composition of the reacting minerals (insets). The amorphous material contains significant concentrations of Na, Mg, and Fe as well as Ca and Al in case of the pyroxene- and amphibole-derived material, respectively. The composition of both materials therefore demonstrates an element transport capacity that is way beyond that of aqueous solutions at temperatures below 800 °C[37].

## Discussion

Our observations outlined above comprehensively demonstrate a so-far unprecedented effective dissolution–reprecipitation process involving an amorphous phase in which alkalis, Al as well as Ca, Mg, and Fe can be effectively transported between reactant and product phases. Our findings unify several processes known from lab experiments with entirely novel observations into a series of naturally occurring processes that differ in several aspects from the common view on mineral reactions. First, an amorphous material forms directly from reacting solids, hence thermodynamically decoupling mineral dissolution from aqueous solutions. The composition of the amorphous material reflects that of the parent mineral, such that alkalis, Al as well as Ca, Mg, and Fe are effectively transported therein. Second, the amorphous material is present in a reaction-induced syn-metamorphic porosity, where it can provide effective transfer of elements necessary for the formation of complex multi-oxide silicates. Third, reaction products can form directly by repolymerization from the amorphous phase, likely facilitated by short-range atomic order effects in the amorphous material. The combination of these mechanisms has crucial implications for all major processes during mineral dissolution, element transport, and reprecipitation within solid materials.

Experimentally determined rates of hydrolysis in geological environments are between $10^{-11}$ and $10^{-14}$ moles/cm$^2$/s[38,39], constraining the duration for the dissolution of mineral grains under natural conditions between $10^3$ and $10^5$ years. The expectance from these data is that natural mineral reaction rates are generally sluggish compared to everyday processes. This is true if the dissolution mechanism follows the "classical" path, i.e., hydrolysis of monolayers and formation of edge pits at the surface of the reacting mineral.

Experimental and natural examples of mineral weathering show, however, that silicate dissolution can involve the formation of a silica-rich amorphous phase[24–27]. The mechanism by which the parent phase dissolution and the formation of an amorphous

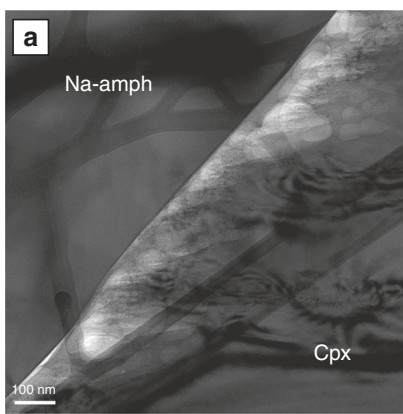

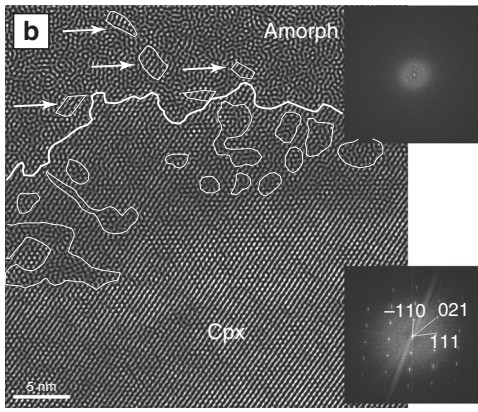

**Fig. 4** Details of the reaction zone between cpx and Na-amph. **a** Bright-field TEM image of the FIB section taken along the red line in Fig. 1c highlighting the denticulated surface of the reacting cpx. The needle-like denticles grade into an amorphous phase by a gradual de-polymerization of the cpx lattice as shown in **b**. **b** Inverse FFT HR-TEM image showing the cpx dissolution process. The cpx surface is highly lobate and within the crystal amorphous regions develop. In turn, small fragments of the original crystal can be seen in the amorphous material (arrows). The calculated diffraction patterns demonstrate the structural differences between the reacting cpx and the amorphous phase

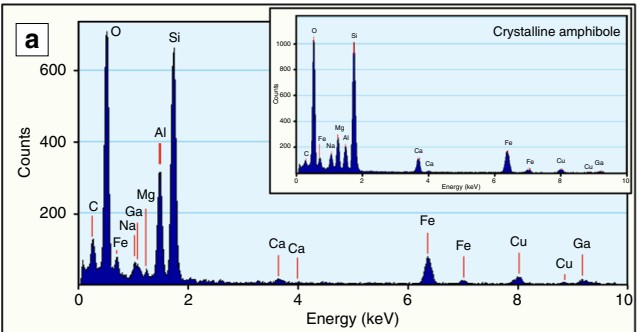

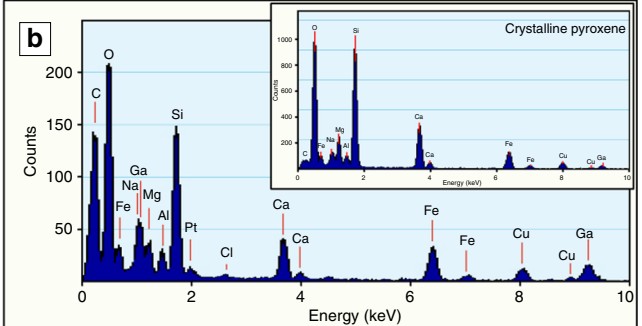

**Fig. 5** Comparison of the energy dispersive X-ray spectra of amorphous and crystalline phases. Comparison is of amorphous material resulting from the dissolution of sodic-calcic amphibole (**a**) and cpx (**b**) with their crystalline source minerals (insets). In both amorphous phases significant amounts of alkalis, Al as well as Fe are present. The expected element transport capacity of these materials is several times larger than that of aqueous solutions. See text for discussion. Ga K and L line intensities are due to Ga ion implantation during FIB milling. Cu K intensity comes from the TEM copper grid. Pt intensity is due to redeposited Pt during FIB milling

phase occurs in these experiments comprises metal-proton exchange[24] and stoichiometric liberation of constituent atoms followed by precipitation of amorphous silica[27]. It has been shown that this process has a fundamental influence on mineral reaction rates[24,27]. On the one hand, it has been argued[27] that an amorphous layer formed during dissolution might actually reduce rather than enhance the reaction rates by shielding the reacting surface. On the other hand, it has further been deduced from such experiments that silicates can precipitate directly by re-polymerization from this amorphous phase, suggesting that a short-range atomic order might exist in the amorphous phases[24]. The latter process, which is observed in our samples (Fig. 2a, b) displays a reaction path, which enables significant enhancement of the dissolution and element transport rate as the limited solubility of silicates in aqueous solutions is no longer controlling the liberation and transport of reacting elements from the reactant.

Other new implications for the element liberation and transport during mineral reactions can be deduced from our observations. In all previous studies where an amorphous phase was observed during mineral dissolution it consisted predominantly of silica or mixtures of alumina and silica[13,24,27], although the dissolving parent solids contained other metals. The reason for that is either incongruent dissolution (leaching) or selective precipitation of amorphous silica[27]. This is in strong contrast to our results. The chemical comparison between parent minerals and amorphous phases in our natural samples (Fig. 5a, b) clearly demonstrates that significant amounts of Al, alkalis and also Ca, Mg, and Fe are liberated from the parent phase and transported within the amorphous material. Regarding the formation process of the amorphous phase, we demonstrate above (Fig. 4b) that depolymerization of pyroxene reacting with an aqueous solvent occurs not only at the outermost layers of the crystal, but proceeds in a region of several tens of nm within the reactant. This observation is supporting the interpretation that selective replacement of metal atoms by proton penetration leads to preferential removal of Ca, Mg, and Fe atoms by protonation of metal–oxygen bonds between the tetrahedral chains. Nevertheless, this mechanism is in contrast to a leaching process as it facilitates the liberation of crystal fragments, such as shown in Fig. 4. Especially in chain silicates, such as pyroxenes and amphiboles, this seems to be a common process[10,24]. This interpretation is also in line with the distinct concentrations of metals in the parent phases and the amorphous products.

In the amphiboles (Fig. 2c), we can further show that the formation of the amorphous material occurs preferentially along crystal defects within reacting grains allowing for dissolution and repolymerization of products even deep within the reactants (Supplementary Note 6). Nano-scale anisotropies, such as the linear dislocation arrays abundant in most of the investigated grains, provide initial pathways for the hydrolysis, indicated by the hydrogen mappings in Fig. 3, and the formation of the amorphous material. It is notable that many partially equilibrated minerals show patchy and irregular compositional zoning patterns, such as finger-like structures (Fig. 1b) emanating from grain boundaries and reaching far into mineral interiors[40]. Konrad-Schmolke et al.[41]. showed similar structures in garnet displaying dislocation arrays or sub-grain boundaries along which element exchange between mineral cores and the grain boundaries is facilitated (Supplementary Figure 11). Hence, reactive fluid fluxes and associated phase transformations can obviously proceed within mineral grains, invoking in situ dissolution and reprecipitation processes.

Another important novel observation is the mobility of the amorphous material as it does not generally precipitate at the reacting surface, and therefore enables direct element transfer

over a distance of at least the width of the pore space shown in Fig. 2a. The amorphous material forms a direct physical link between reactants and product surfaces where precipitation occurs directly by repolymerization from the amorphous phase. This process evades the limited solubility of silicates in a hydrous fluid, which in turn significantly enhances element transport during the reaction.

Regarding the element transfer during the observed mineral transformations the reactions involve predominantly the hydration of clinopyroxene (cpx) and thus the transfer of Na, Mg, Al, and Fe as well as Si and oxygen from the reactants to the product Na-amphibole. It is notable that all of these elements are abundant in the amorphous phase observed in the reaction sites. As the Mg and Fe components in the reacting phases are coupled to Ca in case of cpx ($Ca(Mg,Fe)Si_2O_6$) and NaCa-amphibole ($NaCa_2(Mg,Fe)_4Al_3Si_6O_{22}(OH)_2$) as well as Al in case of garnet ($(Mg,Fe)_3Al_2Si_4O_{12}$), the product side of the reactions must contain an additional Ca or Al phase. In case of the NaCa-amphibole decomposition, the reaction is balanced by the formation of Na-white mica (paragonite), which is abundant in the product assemblage, and the increase in the Ca component in pyroxene, which is reflected in the compositional changes in residual cpx (Supplementary Figure 5). The breakdown reaction of NaCa-amphibole can be written as:

$$\underset{\text{NaCa-amphibole}}{3\ NaCa_2(Mg,Fe)_4Al[Al_2Si_6O_{22}](OH)_2}\ +$$
$$\underset{\text{Na-pyroxene}}{4\ NaAlSi_2O_6}\ +\ \underset{\text{Quartz}}{11\ SiO_2}\ +\ \underset{\text{Water}}{2H_2O}$$
$$=\ \underset{\text{Na-amphibole}}{2\ Na_2(Mg,Fe)_3Al_2[Si_8O_{22}](OH)_2}\ +$$
$$\underset{\text{Paragonite}}{3\ NaAl_2[AlSi_3O_{10}](OH)_2}\ +\ \underset{\text{Ca-pyroxene}}{6\ Ca(Mg,Fe)Si_2O_6} \quad (1)$$

In case of the pyroxene breakdown the reaction involves the consumption of garnet and the formation of Na-white mica:

$$\underset{\text{Na-pyroxene}}{3\ NaAlSi_2O_6}\ +\ \underset{\text{Garnet}}{(Mg,Fe)_3Al_2Si_4O_{12}}\ +$$
$$\underset{\text{Quartz}}{SiO_2}\ +\ \underset{\text{Water}}{2\ H_2O}$$
$$=\ \underset{\text{Na-amphibole}}{Na_2(Mg,Fe)_3Al_2[Si_8O_{22}](OH)_2}\ +\ \underset{\text{Paragonite}}{NaAl_2[AlSi_3O_{10}](OH)_2}$$
$$(2)$$

The observed change in the mineral assemblage during the water influx is allowing for both reactions (1) and (2) as the formation of Na-white mica (paragonite) is observed together with an increase in Ca in the patchy zones of the affected pyroxenes.

The element transfer during these reactions is constrained by the transport capacity of the moderating agent. Experiments and numerical simulations show that the total amount of dissolved solutes (TDS) in hydrous metamorphic fluids below 600 °C is in the order of 30–40 g/kg water[42]. This is an important parameter for all non-isochemical reactions as nutrient cations must be transferred to and from the reaction site. Assuming a solid density of 3 g/cm$^3$, the above values suggest that volumetric fluid–solid ratios in the order of $10^2$ are necessary to precipitate solids from aqueous solutions. The element transport mechanism described here is decoupled from that limited solubility of solids in an aqueous fluid implying that element transport rates can be much higher than previously thought.

Although the composition of the amorphous material could not be precisely quantified (due to the limited thickness of the material in the FIB sections), estimations on the minimum amount of transported material can be made based on electron microprobe analyses. Measured totals in the anhydrous pyroxenes, reflecting the maximum counts obtainable from the TEM foils, yielded about 30 wt% (Supplementary Table 1). In contrast, measurements in the amorphous material yielded only 8.5 wt% in total. Assuming that the difference in the totals results entirely from the hydrous component in the amorphous material, the 8.5 wt% from the amorphous material would correspond to a composition of ~25 wt% solids and ~75 wt% of a hydrous component. Given a minimum amount of 20 vol% of amorphous material in the pores (Fig. 2a), the amount of solid material in this pore would correspond to a TDS value of ~140 g/kg water. This value is almost five times larger than experimental data[42] demonstrating the substantially higher element transport capacity of such an amorphous material. Dissolved elements, such as the presence of Al–Si and alkali-Al–Si polymers in the fluid, would additionally increase this amount[42].

The polymerization of reactants directly from the amorphous material, as obvious in our samples, has been observed in several experiments[24,43,44]. Especially carbonate precipitation seems to be facilitated by amorphous precursor phases. The initial solid products of these experiments are amorphous solids with only short-range atomic order, but it has been shown that the formation of nano-crystalline particles from such amorphous precursor phases facilitates nucleation and growth of crystalline material with a long-range order[27,45,46]. Pan et al.[47], for example, observed the polymerization of crystalline hydroxyapatite after the precipitation of an amorphous calcium phosphate precursor and similar observations have been made during the experimental formation of serpentine[48]. It has also been suggested that the short-range atomic order of the amorphous precursor phases is controlled by the parent mineral structure[27], thus facilitating the transformation of minerals with similar crystal structures. Our observations are supporting this suggestion, as the formation of chained clusters of polymerized material and the precipitation of crystalline solids directly from the amorphous material (Fig. 2b) can only be observed in the NaCa-amphibole to Na-amphibole reaction, where reactant and product have the same crystal structure. In case of the pyroxene-amphibole reaction, where reacting and product phases have different crystallographic properties, a direct repolymerization could not be observed.

Our observations clearly demonstrate the effective generation of the amorphous phase (Figs. 2a and 4) as well as its high element transport capacity. Further, it is obvious that the material allows for direct repolymerization of product phases from the transport agent (Fig. 2b). However, the molecular structure of the amorphous material cannot directly be deduced from the TEM investigations in this work. The material does not show any crystallinity or long-range ordering as indicated by the diffraction pattern shown in Fig. 4b. Nevertheless, it is unclear inasmuch a three-dimensional cross-linked network exists between the molecules in the amorphous substance. The latter would characterize the amorphous phase as a gel. A gel-like structure would also explain the physical connection that is formed by the amorphous material between the reactant and the product (Fig. 2a), which requires a certain degree of physico-chemical bonding within the amorphous material. The high water content of the amorphous phase (Supplementary Table 1) is further supporting a gel interpretation, as a gel is characterized by water surrounding discrete particles or polymer networks.

Inasmuch polymerization and complexation play a role within the amorphous phase is also unclear. It has been shown experimentally that Na–Al silicates can form a wide range of solute complexes[49], which in turn enhance the solubility of (trace) metals by complexation[50]. Such metal complexes occur especially in alkali-Si-rich systems and are most effective element transport agents during mineral reactions[50], but it has never been demonstrated that such polymer- and/or complex-rich fluids exist during naturally occurring dissolution–reprecipitation reactions, nor are their physical properties fully understood. Nevertheless, further investigations at the molecular scale are necessary to undoubtedly identify the physical properties of the amorphous phase observed in our samples.

Nevertheless, all these observations point towards the fact that element transport and reaction rates in natural rocks might be much higher as previously thought. The role of an amorphous transport agent needs to be clarified in future experiments and natural observations. Our results indicate that cluster-attachment-driven crystallization mechanisms, as well as amorphous element transport materials seem to be common in Nature and together with several individual observations in natural rocks[28,29,51,52] demonstrate that the mechanisms described in this paper may be of far greater relevance to dissolution–reprecipitation processes than previously envisaged.

## Methods

**Nano-SIMS.** We measured $^{16}OH^-/^{16}O^-$ and $^{28}SiH^-/^{28}Si^-$ ratios in pyroxenes with a CAMECA nanoSIMS 50L at the Infrastructure for Chemical Imaging at Chalmers University of Technology and University of Gothenburg, Sweden. The measurements were performed with a 16 keV $Cs^+$ primary beam rastering the sample surface. A normal incidence electron gun was used to compensate the charge buildup at the sample surface during the measurements. A 150 pA primary current with a D1-1 diaphragm implanted a dose of $6.2 \times 10^{16}$ Cs cm$^{-2}$ in the sample surface before each measurement. Each measurement consists of a series of image planes of $^{12}C^-$, $^{16}O^-$, $^{16}OH^-$, $^{28}Si^-$, $^{28}SiH^-$ planes acquired simultaneously using a 3 pA $Cs^+$ primary current rastering a $30 \times 30$ $\mu m^2$ window. The primary beam diameter is ~300 nm. Each measurement is a stack of 29–50 planes. Each image plane is made of $256 \times 256$ pixels. The dwell time per pixel is 2 ms. The duration of the analysis ranged from 1 to 1 h 45 min. Mass filtered images were acquired in multicollection mode using the entrance slit 4 (width = 15 μm) and the aperture slit 3 (width = 150 μm). The count rates were measured with electron multiplier detectors in counting mode. The energy slit was kept full size (energy band pass up to 100 eV). The relative transmission of the mass spectrometer is ~26% with a mass resolving power of 10,000 on $^{16}O^-$ (with the CAMECA definition). The $^{28}SiH^-/^{28}Si^-$ and $^{16}OH^-/^{16}O^-$ ratios of a measurement are calculated with the help of the image processing WinImage (CAMECA) software tool. The $^{16}O^-$, $^{16}OH^-$, $Si^-$, and $SiH^-$ counts were extracted in a common region of interest (ROIs hereafter) generated by hand. ROIs area range from 16 to 116 $\mu m^2$. The $^{12}C^-$ images are used to locate and avoid cracks at the sample surfaces for ROIs definitions. Averaged counts rates $^{16}O^-$ and $^{16}OH^-$ range from $1.2 \times 10^5$ to $4.6 \times 10^5$ and from 800 to 3600 counts per second respectively. Averaged $Si^-$ and $SiH^-$ counts range from 4000 to 17,000 and from 100 to 600 counts per second respectively. The count rates were corrected using a deadtime of 44 ns for each detector. The $^{16}OH^-/^{16}O^-$ and $SiH^-/Si^-$ ratios range from $5.4 \times 10^{-3}$ to $8.1 \times 10^{-3}$ and from $2.6 \times 10^{-2}$ to $4.1 \times 10^{-2}$ in the various ROIs defined.

**FIB nanotomography.** The FIB/scanning electron microscope (SEM) nanotomography to visualize the porosity in the internal parts of the pyroxene grains was performed with a FEI Quanta 3D by repetitively sputtering material from the target with the ion beam and then imaging the region of interest with the electron beam. First the ROI was protected by depositing a 2 μm thick layer of Pt on the top of the ROI. Then frontal and lateral trenches were created by sputtering material out from the sample using a focused Ga beam working at an acceleration voltage of 30 kv and 27 nA beam current. Then material was removed from the frontal surface of the ROI using a beam current of 3 nA. For the data acquisition, we used the FIB operating at 30 kV and 1 nA, while the secondary electron images were acquired using accelerating voltage of 20 kV and 300 pA beam current. We have used a continuous thickness of material removal of 50 nm.

**Transmission electron microscopy.** Analytical and energy-filtered high-resolution TEM (HRTEM) using a FEI Tecnai G2 F20 X-Twin at GFZ Potsdam operated at 200 kV with a field emission gun electron source was used for the present study. The TEM is equipped with a postcolumn Gatan imaging filter (GIF Tridiem). The HRTEM images presented were energy-filtered using a 10 eV window on the zero loss peak. Analytical TEM was performed with an EDAX X-ray analyzer equipped with an ultra-thin window. The X-ray intensities were measured in scanning transmission mode, where the electron beam is serially scanned over a preselected area, minimizing mass loss during data acquisition.

**Data availability**. The data sets generated during and/or analyzed during the current study are available from the corresponding author on reasonable request.

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

## Acknowledgements

M.K.-S. and R.H. acknowledge a grant from the German Science Foundation (DFG, KO3750/2-2). M.K.-S. thanks Lutz Hecht (Museum of Natural History Berlin) for access to the microprobe and Thomas Zack (Gothenburg) for pointing out the importance of the investigated rocks regarding fluid–rock interaction processes.

## Author contributions

M.K.-S. has designed the project, took the back scattered electron images, coordinated the contributions from the co-authors, led the discussion about the data interpretation, and wrote most of the text. R.H. and R.W. contributed to the discussion and data interpretation and wrote parts of the text. R.W., A.T., L.M., and F.D.H.W. were responsible for the TEM, Nano-SIMS, nanotomography, and electron microprobe analyses, respectively, as well as contributed to the interpretation of the respective data. A.S. was responsible for the FIB sectioning. N.K. performed the petrographic investigations of the samples.

## Additional information

**Competing interests:** The authors declare no competing interests.

