## [Peer Review File · Nature Communications]

Reviewers' comments:

Reviewer #1 (Remarks to the Author):

Detailed analytical studies of this kind are to be welcomed as a way of advancing our understanding of fluid-mineral interaction in rocks and the mechanisms of dissolution-reprecipitation reactions. The standard of characterization of the reactions described in this paper is excellent, and the authors use this to argue that they have found a new mechanism that in some way deviates from the "classical dissolution-transport-precipitation path" (line 54). Critical to their case is the interpretation of the role of the amorphous phase.

However I find the interpretation in this paper confusing and at times even contradictory.

In describing the classical model, the authors state that element exchange occurs via "ionic solutions" (lines 16 and later). While reactions such as in Ref 4 do take place in ionic solutions (KBr-KCl) this is not the case in general, and in Refs 1-3 as well as other references to this mechanism, ionic solutions are not specified, in fact coupled dissolution-precipitation has also been described within silicate melt as the fluid phase.

Any natural aqueous solution in a rock will contain many dissolved species and in the case of silica saturated solutions is likely to be partly polymerised, not ionic.

The abstract states that the amorphous material forms directly by depolymerization of the crystal lattice (line 20), but in line 57,58 the solid amorphous material forms by dissolution of the primary minerals. I assume that this means from the dissolved material. This is confusing, especially when later the authors cite Hellmann (ref 28) and Casey (ref 29) which seems to be quite a different situation and controversy. More on that below.

In the classical case described by the authors, a fluid infiltrates a rock that is out of equilibrium (in their case the eclogite), and begins to dissolve the mineral phase(s) in the eclogite. From this interstitial fluid (whose composition reflects a mix of eclogite composition with the composition of the infiltrating fluid), minerals belonging to blueschist facies crystallise. In natural rocks the remaining fluid may be partly incorporated in any hydrous phases that form (in their case amphibole) but may also migrate out of the rock, or be trapped.

The description of the microstructures in this paper and the distribution of the amorphous phases would be consistent with the well-known mechanism previously described, if one assumed that the amorphous phase represents the desiccated interstitial trapped fluid. The rest of the description in the paper would be consistent with this (e.g. "that the amorphous material can move freely within the pore space" line 180). The 'depolymerization' of the amorphous material could also indicate the early stages of reprecipitation from a fluid phase.

The authors should make a case why that is not a possibility.

I do not see the relevance of ref. 25 especially to argue that the amorphous material has an element transport capacity way beyond ionic solutions (line 190). Incidentally, do the authors really believe that fluids in these rocks are "ionic" ?

I think it is difficult to infer much from the composition of the amorphous phase, more so given the difficulty in obtaining a good analysis. If, as is stated in the abstract, the amorphous phase somehow forms directly from the parent phase by depolymerization, then it might be expected that its composition would at least approximate to the parent composition, which it does not. On the other hand, during dissolution there might be, at some time point, a solution which is a mix of parent and incoming solution composition from which a new mineral phase will crystallize, leaving a fluid which contains the "unused" elements and may be trapped. Isn't this more consistent with

the measured composition?

The references to the amorphous phases described by Casey and then Hellmann seem to be entering quite a different controversy. When silicates dissolve, it is common to find that they are replaced by amorphous silica. Casey argued that this was because the other metal components were “leached out” by preferential dissolution. Many papers since Casey (including by Hellmann) have shown using isotopic and other analytical methods that the silica replacement is more likely to be due to a coupled dissolution – reprecipitation mechanism whereby the silicate mineral dissolved congruently and from this solution, amorphous silica precipitated (initially as a gel before desiccation – e.g. King et al Am Min 2011). It has generally been argued (e.g. Daval et al. Chem. Geol. 2011) that such silica rims may represent a barrier to further dissolution as element transport is more limited compared to that in solution i.e. opposite that argued here.

Without explicitly explaining the Casey and Hellmann references and merely quoting them as examples of amorphous phase formation, I do not see how they contribute to the case made in this paper.

Overall, I do not see that the authors have made a case for a “novel mineral reaction mechanism” that has “major implications”. The role of amorphous material during replacement reactions and crystal growth is a complex topic (even calcium carbonate can form from aqueous solution via an amorphous precursor phase. E.g. Rodriguez-Navarro et al. Crystal Growth & Design, 2016) and I am not convinced that by documenting the existence of amorphous phases in a transforming rock, the authors have found a novel mechanism.

The paper, together with the very lengthy supplementary material that has very interesting replacement microstructures, might be better presented as a longer paper rather than aiming for a higher profile by overstating the novelty of their observations.

Reviewer #2 (Remarks to the Author):

The paper presents evidence from a natural occurrence that reactions between solids (minerals in this case) occur via the mediation of an amorphous fluid like phase rather than an aqueous ionic solution. The main implication of this observation is that reactions can occur much faster and by a different mechanism than what is generally assumed. The observation is novel. As stated in the paper, the results should be of interest in a wide range of fields where solid state reactions are considered, quite beyond geochemistry or the mineral sciences (e.g. materials science). The observation should considerably influence and reorder thinking on how solids react with one another. The observation has been carefully documented by different kinds of high quality TEM measurements and analytical data from nanosims. I think the claims are adequately documented, and previous literature has been suitably cited. The manuscript is clearly written and concise and I feel it should be published.

The only thing that I feel would enhance the paper is naming the reaction, and a discussion of it: What exactly is the reaction for which the mechanism is being observed and the mechanism is being discussed in such detail? This information should be available from the phase diagrams for the rock. This would also help readers to evaluate how commonly the reaction is actually encountered in nature (i.e. is this a one-off situation?) and exactly what components show an open system behavior.

Also, while it is established that an amorphous intergranular phase functions as a transport

medium, it has not been demonstrated that the material was a "gel". It may be better to avoid that, particularly in the title, and just stick to "amorphous phase".

In the following we demonstrate how we changed our manuscript according to the reviewers' suggestions and comments. We thank both reviewers for the thorough and critical review and improved the manuscript along their comments. As our revised manuscript contains some important changes as requested by the reviewers we additionally submit an annotated manuscript together with the new text in order to demonstrate our changes in detail. We think that, based on the reviewers suggestions, the new manuscript reads much better and explains in more detail our observations and interpretations.

We have added Anja Schreiber (GFZ Potsdam) to the author list as she significantly contributed to the success of this research with her excellent skills in producing electron transparent FIB slices. Without her the observations made in this project would have never been possible.

In the following reviewers' comments are in italics, our comments are in regular face.

Reviewers' comments:

Reviewer #1 (Remarks to the Author):

Detailed analytical studies of this kind are to be welcomed as a way of advancing our understanding of fluid-mineral interaction in rocks and the mechanisms of dissolution-precipitation reactions. The standard of characterization of the reactions described in this paper is excellent, and the authors use this to argue that they have found a new mechanism that in some way deviates from the "classical dissolution-transport-precipitation path" (line54). Critical to their case is the interpretation of the role of the amorphous phase.

We appreciate that reviewer 1 thinks that the work is of broad interest and that the quality of the presentation has a high standard.

However I find the interpretation in this paper confusing and at times even contradictory.

We have changed parts of the text in order to make our points clearer to the reader. In the following we will show in detail how we improved the manuscript along the reviewers' suggestions.

In describing the classical model, the authors state that element exchange occurs via “ionic solutions” (lines 16 and later). While reactions such as in Ref 4 do take place in ionic solutions (KBr-KCl) this is not the case in general, and in Refs 1-3 as well as other references to this mechanism, ionic solutions are not specified, in fact coupled dissolution-precipitation has also been described within silicate melt as the fluid phase.

Any natural aqueous solution in a rock will contain many dissolved species and in the case of silica saturated solutions is likely to be partly polymerised, not ionic.

We apologize for the simplified and insufficient description of the solution that serves as transport agent during mineral reactions. Stating that solutions are general ionic was not entirely correct as there are other forms of element and molecule transport in aqueous solutions possible. Furthermore, we are aware of the fact that coupled dissolution-precipitation reactions also occur in silicate melts. However, we changed the respective paragraphs and generally avoid the statement that solutions are generally “ionic”. We now state that in most cases of fluid mediated mineral reactions the transport agent is an “aqueous fluid”.

The abstract states that the amorphous material forms directly by depolymerization of the crystal lattice (line 20), but in line 57,58 the solid amorphous material forms by dissolution of the primary minerals. I assume that this means from the dissolved material.

This is a misunderstanding due to an unclear description from our side. As we now state in the revised text in both instances as well as in the Figures we can show that the amorphous phase forms directly by depolymerization of the primary mineral. In the second case mentioned by the reviewer we used the word “dissolution” instead of “depolymerisation”, which we have now corrected. Regarding the process by which our investigated elements dissolve we are in the revised version much more precise regarding our wording.

This is confusing, especially when later the authors cite Hellmann (ref 28) and Casey (ref 29) which seems to be quite a different situation and controversy. More on that below.

We believe that this confusion caused the concern mentioned above and, as we have clarified in the revised version what we mean, we think that the potential controversy is also clarified.

In the classical case described by the authors, a fluid infiltrates a rock that is out of equilibrium (in their case the eclogite), and begins to dissolve the mineral phase(s) in the eclogite. From this interstitial fluid (whose composition reflects a mix of eclogite composition with the composition of the infiltrating fluid), minerals belonging to blueschist facies crystallise. In natural rocks the remaining fluid may be partly incorporated in any hydrous phases that form (in their case amphibole) but may also migrate out of the rock, or be trapped.

We agree to this description of rock metamorphism. However, the point that we make in our contribution is that (1) the formation of the transport agent, in our case an amorphous material, involves hydrolysis that penetrates into the crystal lattice in a region several nanometers below the surface, (2) the amorphous material reflects the composition of the parent phases and has therefore much higher element concentrations than classical solutions and (3) the newly formed phases precipitate directly from the amorphous material by forming polymerized chains that are continuously attached to the crystal surfaces. The observation that these mechanisms of dissolution, transport and re-precipitation that are clearly different from the classical processes occur in natural rocks make our findings unique and new. Several aspects of our observations have been deduced from experimental works. The

circumstance that we demonstrate for the first time the entire process of an amorphous phase-mediated dissolution-precipitation process actually occurring in Nature, together with our new findings, such as the composition of the amorphous phase and its transport capacity, make our observation novel and important for a broad audience.

The description of the microstructures in this paper and the distribution of the amorphous phases would be consistent with the well-known mechanism previously described, if one assumed that the amorphous phase represents the desiccated interstitial trapped fluid.

Here are several of our major points of our TEM observations addressed that demonstrate the formation and properties of the amorphous phase. From these observations it seems very unlikely that the amorphous phase results from desiccation of a conventional metamorphic fluid:

- We can show in Figures 2 and 4 that the amorphous material forms directly by depolymerisation of the parent mineral. Further, our detailed observations also show that the amorphous material occupies large parts of the pore space, and as we can also show that the composition of the amorphous material contains elements other than Si, such as Ca and Fe, is capable of transporting significantly larger amounts of elements than conventional fluids.
- Furthermore, we can clearly show in Fig. 2A that the amorphous material is emerging from the parent mineral and also in contact with the new phase, where it repolymerizes directly to form the precipitate. The material displays a physical connection between parent and product phase. We do not understand why this could be interpreted as a conventional solution mediated dissolution-precipitation. We believe that this is evident from Figs. 2 and 4, where the entire process is visualized and clearly different from the conventional model.
- Regarding the desiccation: We assume that our text might have been a bit unclear in this regard. We do think that there is a fluid present during the entire process and the amorphous material is contained in this fluid phase. The situation that we observe here is, of course, after the fluid has left. However, our point in this paper is that we can visualize the formation of the amorphous material (Figs. 2C and 4B), we see the material communicating between the reactant and product (Fig. 2A) and we can demonstrate the repolymerization process (Fig. 2B). Additionally, we demonstrate with the EDX analyses that the composition of the amorphous material is significantly different from the composition of a conventional solution, which is supported by our – although simplified – conservative constraints on the amount of transported elements. Thus, we demonstrate the formation, transport capacity and repolymerization of the amorphous material, all of which processes are different from the classical model, therefore we do not fully understand inasmuch our observations should reflect the classical path. The combination of the observations in Figs. 2, 4 and the EDX analyses rule out that possibility. All the processes visualized in this paper and highlighted in this rebuttal show that the amorphous phase cannot be simply the desiccated material from a classical solution, its formation, its composition and the behaviour at the product surface show that!

In order to address these points we have added the paragraph “Molecular properties of the amorphous material” in which we discuss different possibilities of the molecular structure within the mediating amorphous material.

The rest of the description in the paper would be consistent with this (e.g. “that the

amorphous material can move freely within the pore space” line 180). The ‘depolymerization’ of the amorphous material could also indicate the early stages of reprecipitation from a fluid phase.

The depolymerisation of the reactant is demonstrated in the high resolution TEM images in Fig. 4. As the cpx is the reactant, “depolymerisation” cannot represent early stages of the reprecipitation. However, we also have to take into account that reviewer 1 apparently confused de- and repolymerization and with this comment presumably refers to the formation of the new phase shown in Fig. 2A. This is an assumption as the statement of the reviewer is not clear at that point. However, also the precipitation process, which we could trace with our high resolution TEM images, is clearly different from the classical approach. The HR TEM profile shows that the re-polymerization occurs first by the formation of chains that have a slightly higher distance between them as in the later amphibole. These chains represent the first stages of re-ordering of the amorphous material. This is in strong contrast to the classical precipitation model, where single atoms from the aqueous solution are added to monolayers at the crystal surface.

The authors should make a case why that is not a possibility.

We think that in the revised version we are much clearer about that.

I do not see the relevance of ref. 25 especially to argue that the amorphous material has an element transport capacity way beyond ionic solutions (line 190). Incidentally, do the authors really believe that fluids in these rocks are “ionic” ?

This comment is a follow up of the statement about ionic solutions above and we admit that the description of solutions in our text was too simplistic. We have changed that in the revised version. However, this does in no way influence our observations and interpretations, as in the classical model, formation of the transport agent (ionic or not) is different (depolymerisation of monolayers), the solution itself is not an amorphous phase nor has it shown to consist to large parts of such an amorphous phase as shown in Fig. 2A, nor does the classical model assume repolymerization from an amorphous phase.

I think it is difficult to infer much from the composition of the amorphous phase, more so given the difficulty in obtaining a good analysis. If, as is stated in the abstract, the amorphous phase somehow forms directly from the parent phase by depolymerization, then it might be expected that its composition would at least approximate to the parent composition, which it does not.

We do entirely disagree with this statement and we are a little bit confused how the reviewer can oversee the significant amounts of Fe and Ca that are undoubtedly contained in the material that formed from the cpx (Fig. 5). In both cases, formation from amphibole and cpx, the dissolved material indeed reflects that of the parent crystal, this is one of the major observations! We have added EDX spectra of the parent minerals in order to better demonstrate that the amorphous material indeed resembles the composition of the host material.

On the other hand, during dissolution there might be, at some time point, a solution which is a mix of parent and incoming solution composition from which a new mineral phase will crystallize, leaving a fluid which contains the “unused” elements and may be trapped. Isn't this more consistent with the measured composition?

As we state in the revised text the composition of the amorphous material does not reflect potential compositions of classical solutions.

The references to the amorphous phases described by Casey and then Hellmann seem to be entering quite a different controversy. When silicates dissolve, it is common to find that they are replaced by amorphous silica. Casey argued that this was because the other metal components were “leached out” by preferential dissolution. Many papers since Casey (including by Hellmann) have shown using isotopic and other analytical methods that the silica replacement is more likely to be due to a coupled dissolution – reprecipitation mechanism whereby the silicate mineral dissolved congruently and from this solution, amorphous silica precipitated (initially as a gel before desiccation – e.g. King et al Am Min 2011). It has generally been argued (e.g. Daval et al. Chem. Geol. 2011) that such silica rims may represent a barrier to further dissolution as element transport is more limited compared to that in solution i.e. opposite that argued here.

As we stated above we can demonstrate that the amorphous material is not only (Al-bearing) silica, but rather contains significant amounts of other metals (e.g. Ca, Na, Fe, Mg). Fig. 2A clearly shows that the material is capable of transporting these elements to the reactant therefore enabling an effective element transport. However, if the amorphous material precipitates at the surface of the reacting mineral, such as shown in the appendix Fig. A8, it might also form a barrier for further reaction progress. This is not in contrast to the new reaction mechanism that is described in this manuscript. We have addressed this point in the revised manuscript in the paragraph “*Implications for crystal dissolution and element transport*”

Without explicitly explaining the Casey and Hellmann references and merely quoting them as examples of amorphous phase formation, I do not see how they contribute to the case made in this paper.

We think that both works significantly contributed to the advancement of non-classical reaction mechanisms and that they demonstrate in experimental studies similar processes that we observe in our natural rocks: (1) We cite Casey predominantly for the observation of the formation of repolymerized silicates, a process that we also observe in our natural samples. (2) We cite Hellmann et al as they formulate the conjecture that crystallographically “alike” phases might repolymerize directly from an amorphous phase, a process that we show to exist in Nature (our Fig. 2A). In the revised manuscript we better integrate these works into the discussion and clarify inasmuch they make important observations relevant for the reaction- and transport rates of mineral reactions.

Overall, I do not see that the authors have made a case for a “novel mineral reaction mechanism” that has “major implications”. The role of amorphous material during replacement reactions and crystal growth is a complex topic (even calcium carbonate can form from aqueous solution via an amorphous precursor phase. E.g. Rodriguez-Navarro et al. Crystal Growth & Design, 2016) and I am not convinced that by documenting the existence of amorphous phases in a transforming rock, the authors have found a novel mechanism.

We entirely disagree with this statement. We state explicitly in our text and more precisely in the revised version that several aspects of the here observed recrystallization process have been observed before in experiments, such as the amorphous precursor phases during carbonate precipitation, and we correctly cite this literature, which might have been overseen by reviewer 1. No publication has so far shown the entire cycle of dissolution, transport and re-precipitation following a process described in detail here. We think that this is made clear in our visualizations and in the referencing, in which we refer to previous works that concentrate on different aspects of this new mineral reaction mechanism. We apologize if we

forgot the above-mentioned references (which we added to our new reference list), but these do only complement our reference list and do not point to an aspect that we have overseen.

The paper, together with the very lengthy supplementary material that has very interesting replacement microstructures, might be better presented as a longer paper rather than aiming for a higher profile by overstating the novelty of their observations.

We cannot understand why the reviewer assumes our observations are oversold: We think we have shown in our rebuttal above that the observation made in this contribution are not in agreement with processes during the classical reaction model. No publication has shown this mechanism to be active during mineral reactions, neither in general, nor in such detail. Our contribution to the understanding of mineral reaction is that we clearly visualize every single step of a reaction mechanism involving an amorphous phase that is capable of transporting significant amounts of metals and allows for direct re-polymerization of product phases. This has not been shown for the classical model, where the transport agent typically cannot be observed as it has left the system. If the aqueous fluid leaves, the transport agent is gone in case of the classical model. In our case, the transport agent is still observable as it differs in its properties from that in classical models. Furthermore, there is a wealth of information regarding reactive fluid migration (the formation of the amorphous phase along the dislocation cores, the formation of an interconnected pore space, etc.) and general observations that might be interesting for material science (as emphasized by reviewer 2) and a wide range of geosciences (including geochemistry, ore geology and geodynamics) that publication in a more specialized journal would hide this information from potentially interested readers.

Regarding the appendix, we think that all necessary material to support our interpretations is in the main text and the appendix contains only material that is interesting for those who want to have a better insight into the geology, petrology, chemistry and microstructures of the investigated rocks.

Reviewer #2 (Remarks to the Author):

The paper presents evidence from a natural occurrence that reactions between solids (minerals in this case) occur via the mediation of an amorphous fluid like phase rather than an aqueous ionic solution. The main implication of this observation is that reactions can occur much faster and by a different mechanism than what is generally assumed. The observation is novel. As stated in the paper, the results should be of interest in a wide range of fields where solid state reactions are considered, quite beyond geochemistry or the mineral sciences (e.g. materials science). The observation should considerably influence and reorder thinking on how solids react with one another. The observation has been carefully documented by different kinds of high quality TEM measurements and analytical data from nanosims. I think the claims are adequately documented, and previous literature has been suitably cited. The manuscript is clearly written and concise and I feel it should be published.

The only thing that I feel would enhance the paper is naming the reaction, and a discussion of it: What exactly is the reaction for which the mechanism is being observed and the mechanism is being discussed in such detail? This information should be available from the phase diagrams for the rock. This would also help readers to evaluate how commonly the reaction is actually encountered in nature (i.e. is this a one-off situation?) and exactly what components show an open system behavior.

We agree with the reviewer's statement that we did not stress the importance of the observed reaction. Therefore, we added a paragraph in the introductory part stating:

“It is notable that the re-hydration of eclogites and the formation of hydrous amphibole from pyroxene is a commonly observed fluid-rock interaction in high-pressure rocks (Hacker et al., 2003). The reaction is associated with a significant positive volume change and the consumption of fluids percolating in and above subducted plates. Hence, it has crucial implications for the seismicity and melt production in subduction zones (Peacock et al., 2011; Hacker et al., 2003).”

We also added a paragraph to the discussion explaining in detail the observed mineralogical changes and how the observed reactions can be stoichiometrically balanced. We tied the balanced reaction to the discussion about the transport properties of the amorphous material and the comparison with literature data.

Furthermore, we added Fig. A2 and a text paragraph to the electronic supplementary material in order to better demonstrate the geophysical and geochemical implications of our findings.

Also, while it is established that an amorphous intergranular phase functions as a transport medium, it has not been demonstrated that the material was a “gel”. It may be better to avoid that, particularly in the title, and just stick to “amorphous phase”.

We changed the term “gel” in the revised manuscript and now speak of an “amorphous phase”. We also think that this reflects more general our findings and does not imply that we can undoubtedly show that the amorphous phase is a gel. Additionally, we discuss the molecular properties in the newly added paragraph and make clear, why we think that the amorphous material has gel-like properties.

We hope that we have sufficiently addressed all of the points that were raised by the reviewers and believe that the revised version of this manuscript clarifies all our statements. We again thank the reviewers for their thorough work and for the critical but constructive comments that helped to significantly improve the manuscript.

We are of course open to discuss and clarify potentially open points that are not sufficiently clear.

Reviewer #1 (Remarks to the Author):

The authors have significantly clarified the text and satisfactorily responded to the reviewers' comments. Irrespective of whether the interpretation* of the significance of the amorphous phase turns out as the authors claim, the excellent quality of the analytical and structural characterisation of this important aspect of fluid-rock interaction warrants publication.

* The asterisk refers to the fact that Hellmann among others have described the amorphous phase formed during experimental dissolution of feldspars as a reprecipitated phase, rather than a directly depolymerised phase (as claimed by Casey et al). If this is still controversial then the present paper does not enter into that discussion - they bundle Hellmann and Casey together to reference amorphous phases formed during reaction.

Reviewer #1 (Remarks to the Author):

The authors have significantly clarified the text and satisfactorily responded to the reviewers' comments. Irrespective of whether the interpretation* of the significance of the amorphous phase turns out as the authors claim, the excellent quality of the analytical and structural characterisation of this important aspect of fluid-rock interaction warrants publication.

* The asterisk refers to the fact that Hellmann among others have described the amorphous phase formed during experimental dissolution of feldspars as a reprecipitated phase, rather than a directly depolymerised phase (as claimed by Casey et al). If this is still controversial then the present paper does not enter into that discussion - they bundle Hellmann and Casey together to reference amorphous phases formed during reaction.

Response:

We have deleted the Hellmann et al., 2012 reference after the sentence:

“On the other hand, it has further been deduced from such experiments that silicates can precipitate directly by re-polymerization from this amorphous phase, suggesting that a short-range atomic order might exist in the amorphous phases²⁴”.

We agree with the reviewer's statement that only Casey et al., 1993 (reference 24) can undoubtedly show direct re-polymerization of the amorphous phase. Hellmann et al., 2013 only suggest that a short-range order might exist in the amorphous phase (page 212, second paragraph), but in their case the observed amorphous substance is clearly a reprecipitate. Our interpretations are not influenced by skipping this citation.